# SmartGroup: A Tool for Small-Group Learning Activities

Haining Zhu [1], Na Li [2], Nitish Kumar Rai [2] and John M. Carroll [2,*]

1 Stack Exchange Inc., New York, NY 10038, USA
2 College of Information Science and Technology, Pennsylvania State University, State College, PA 16803, USA
* Correspondence: jmcarroll@psu.edu

**Abstract:** Small-group learning activities (SGLAs) offer varied active learning opportunities and student benefits, but higher education instructors do not universally adopt SGLAs, in part owing to management burdens. We designed and deployed the SmartGroup system, a tool-based approach to minimize instructor burdens while facilitating SGLAs and associated benefits by managing peer group formation and peer group work assessment. SmartGroup was deployed in one course over 10 weeks; iterations of SmartGroup were provided continuously to meet the instructor's needs. After deployment, the instructor and teaching assistant were interviewed, and 20 anonymous post-study survey responses were collected. The system exposed students to new perspectives, fostered meta-cognitive opportunities, and improved weaker students' performances while being predominantly well-received in terms of usability and satisfaction. Our work contributes to the literature an exploration of tool-assisted peer group work assessment in higher education and how to promote wider SGLA adoption.

**Keywords:** active learning; collaborative learning; peer assessment; peer evaluation; peer grading; appeal; team-based learning; teamwork; education technology; iterative design; meta-cognition





## 1. Introduction

In traditional instructor-centered classrooms, the instructors are regarded as the only disciplinary experts that work as knowledge providers while students act as passive recipients of information [1]. Such a one-directional knowledge transmission mode in classrooms decreases students' participation and motivation to learn and is unsuitable for cultivating students' critical thinking [2]. To reverse the situation, previous research examined how to promote students' active learning (i.e., where students engage actively in their learning instead of passively) through engaging them in small group learning activities (SGLA) [3,4], whereby small groups of students work together to meet shared learning goals [5].

Compared with traditional didactic teaching methods, small group learning activities (SGLA) provides experiential learning from which students' collaboration skills are developed not only in academic performance but also in their future careers. Experiential team-based work also helps students develop verbal communication skills, fosters their problem-solving abilities, and improves interpersonal relations skills [6] by promoting elaboration skills and the appropriateness of students' responses to peers seeking assistance [7]. When learners work together on tasks that are meaningful to them, they naturally describe, explain, listen, and interpret, thus developing language skills, collaboration skills, and self-monitoring or meta-cognitive skills through group rehearsal (i.e., learning from teaching peers) [8]. Shared knowledge-building thus allows learners to integrate creation and reception, to negotiate meaning and purpose, to divide and manage collective work, and to come to regard themselves as persons who solve problems and develop conclusions. This outcome appeals keenly to future employers of university students [9].

Even though this article describes an SGLAs research study that was carried out before the COVID-19 pandemic happened, the virtual paradigm of online teaching and

learning in higher institutions during the pandemic adds more importance to exploring the potential of SGLAs in improving students' interactions with peers, learning engagement, and motivation. Since the year 2020, the sudden outbreak of the COVID-19 pandemic forced most higher institutions to transit to "emergency synchronous online instruction" [10] in the U.S. Previous studies showed that emergency remote education gave rise to some new challenges for instructors and students in comparison with in-person education. For instance, students became more demotivated or disengaged in online classes [11,12], and easily become bored or fatigued by Zoom [13] after staring at screens for an extended period of time without personal interaction [14]. In addition, virtual education makes students feel isolated and lack connection to instructors and peers [15] without physical interactions that would otherwise happen in classrooms [11]. Even though most universities have shifted back to in-person classes in the year 2022 in the U.S., hybrid education that combines synchronous online classes with in-person classes has become a new normal of education in the post-pandemic era [16]. Because students' learning engagement is closely related to their interactions and collaboration with peers or faculty [17], this new education trend (hybrid education) in turn makes it necessary to explore the potential of SGLAs in offering students opportunities to collaborate with each other, thereby improving their learning experience especially while they are taking online classes.

Despite all the aforementioned benefits, SGLAs are underutilized in higher education, resulting from a lack of coherent instructional design, collaborative activity design, and technology support for SGLAs [18,19]. In addition, SGLAs would lay extra burdens on instructors who should undertake different responsibilities including class administration, group facilitation, and oversight. Such burdens make it more intimating and challenging for instructors to adopt SGLAs in classrooms. Therefore, if SGLAs are going to be effectively integrated into instructors' practices, we must reduce instructor burdens for adopting this collaborative learning technique. Although previous studies showed that some learning management systems have been utilized to facilitate SGLAS, such as Moodle, on which collaborative learning activities including forums, vocabulary, and databases could be done by students [20]; Canvas [21]; or other more mundane tools such as Media Wiki, Google docs [22], and Blogs [23], none of the tools could reduce instructors' burdens by automating instructors' workflow activities such as grading, managing student groups, etc. Even though the Moodle workshop tool [24] and Canvas [21] can enable peer assignment reviews, they cannot enable students' flexibility to change groups or conduct appeals if they are unsatisfied with the peer assessments.

To facilitate students' collaborative learning and reduce instructor burdens, we developed and deployed the SmartGroup system, an SGLA class management tool; the guiding idea behind the system is to use technology to foster collaborative interactions between students, and automate workflow activities (e.g., forming and managing groups, and grading) with potentially no apparent drawbacks [25]. SmartGroup reduces instructor burdens by allowing them to divide a class into small teams and direct these teams to carry out a learning activity. Most importantly, the system specifically allows students to deliver a project to peer graders, receive feedback from peer graders, and optionally appeal a grading to other peer graders. SmartGroup thus can programmatically create and manage a thread of peer-accountable collaborative learning activities.

We deployed the SmartGroup system at a public research university located in the U.S. Mid-Atlantic region for the fall 2018 semester in a course (N = 42; 1 instructor, 1 TA, 40 students) titled organization of data. After the deployment, we interviewed both the instructor and the TA, and collected 20 anonymous survey responses. During deployment, our minimalistic initial iteration of SmartGroup was continuously redesigned and deployed to meet the instructor's needs; a key theme for iterations was to provide the instructor with greater control over SGLAs and management (grouping, grading, etc.). Iterations are recorded and discussed chronologically. Our results reveal findings consistent with the literature regarding the benefits of peer individual work assessments and SGLAs, and we highlight interesting cases throughout our deployment. We end with a discussion of

these findings and implications regarding promoting wider SGLA adoption. This work is designed to offer contributions specifically to education technology. Education technology and group collaboration research in CSCW are becoming increasingly prevalent, and this work adds to these growing bodies of literature. Specifically, we offer the CSCW community exploration of a tool-based approach aimed at reducing instructor burdens while facilitating quality SGLA learning for students in a more scalable fashion. We hope this tool-based approach can effectively lead to the management of scale-free SGLA. This work's main contributions are:

1.  Descriptions of the tool ensemble as an instructional intervention, its iterative design process, and rationale for iterative changes.
2.  An empirical study of tool-assisted, student-managed, and student-evaluated peer group work in SGLAs, including classroom experiences and peer assessment and grouping consequences.
3.  A discussion of future opportunities for active learning, meta-cognitive benefits and effects of peer group work assessment, and how to further promote SGLA adoption in higher education.

## 2. Related Work

In the following section, we will briefly review the literature regarding the philosophical background of SGLAs, active learning, peer assessment, grouping technologies and tools.

SGLAs are an implementation of the concept of experiential education: that authentic doing, and reflection on doing are primary paths to learning [26,27]. Constructivist learning is a process of enculturation, constituted by the appropriation of the artifacts and practices of a community through collaboration, social norms, tool manipulation, domain-specific goals and heuristics, problem-solving, and reflection in action directed at authentic challenges, that is, challenges that are consequential to the learner [28–31]. In this, collaborative learning through group activity is essential; mentoring, coaching, and cooperation are primary learning mechanisms. Learners develop their own understandings through articulating and demonstrating knowledge to other learners, as well as benefiting themselves from the assistance of other learners [32,33]. SGLAs are thus one type of collaborative learning activity that facilitates active learning.

### 2.1. Active Learning Technologies and Techniques

Technology-facilitated active learning has many fruitful research avenues (e.g., classroom response systems [34], blended classrooms [35], minute papers [36], flipped classrooms [37], and online synchronous peer learning frameworks [38]). During the COVID-19 pandemic, to promote students' active learning in online settings, different learning management tools, and mobile devices have been employed by higher institutions [39]. For instance, classroom response systems (clickers) have been adopted to allow students to vote or to answer simple choice questions posed by instructors [40] in synchronous online classrooms. Although such technology system increased the interactions between students and faculty, it also results in limited student activity and engagement, literally reducing the students' participation to a button push. Moreover, blended classrooms integrate online exercises or activities into a classroom presentation [35]; these approaches involve students more actively, though only within an a priori interaction space. "Minute papers" and other small, lightly graded or ungraded activities allow students to quickly analyze or practice a particular point or operation [36]; such interventions are highly engaging and involve students creatively, but students frequently get no direct feedback. Flipped classrooms [37] have students encounter and study information outside class, and then subsequently apply and practice concepts and skills within class meetings (working either as individuals or teams). Flipped classrooms turn class meetings into workshops, which may be engaging, the instructor (and teaching assistants) are still a bottleneck, as they can only consult with

students (or groups of students) one at a time. Our work explores the use of technology to reduce this bottleneck and provide students with opportunities for active learning.

### 2.2. Peer Assessment

Our approach to reducing instructor burdens in the adoption of SGLAs and collaborative learning relies upon utilizing students and self-guided learning and instructional resources through group work and peer evaluation. Peer assessment can have positive effects on student attitudes and outcomes which are as good as or better than instructor assessments [41], but the major strength of peer assessment is that it allows students to receive quality and timely feedback regarding projects in a manageable way for lecturers [42]. Such feedback is often available in greater volumes, and because of potentially greater amounts of time, peer assessors of lesser assessment skills are still capable of producing feedback on par with instructors [43]; student peer assessment feedback can more closely approximate instructor feedback if students are aware of clearly defined assessment criteria [44]. Although researchers have thought that student involvement (e.g., devising the scheme for intervention and training in its use) is necessary for students to benefit, students feel that they benefit from peer assessment interventions even without said involvement [45]. These benefits come from both the student's roles as an assessor and as an assessee [46]. We note that the benefits of peer assessment complement the benefits of SGLAs in higher education.

Technologically facilitated peer assessment is a promising avenue for supporting peer assessment in group work; peer evaluators in such technologically assisted systems still receive benefits from both their roles as an assessor and an assessee [46]. For example, the peer assessment platform PeerStudio uses technology to facilitate rapid rubric-based feedback for individuals' works and offers students grades based on statistical manipulations of peer reviews; this platform demonstrates that students in large classrooms (i.e., both online and in-person) can benefit from such systems in the form of improved outcomes [47]. Prior research by Kulkarni et al. shows that peer feedback in such online classrooms that utilize rubric-based assessment correlates highly with staff-determined grading [48]; however, such use of rubrics in peer evaluations creates significant time burdens on students, which is why hybrid grading approaches (i.e., machine grading/algorithmic scoring in conjunction with peer rubric-based evaluations) might be favorable [49]. Such hybrid grading systems also allow instructors to offer students questions with free responses, instead of just multiple choice questions [50]. In addition, researchers have developed tools intended for the evaluation of peers' contributions within group projects such as CATME [51]. CATME (i.e., Comprehensive Assessment of Team Member Effectiveness) is a prominent tool that offers both peer evaluation and group formation capabilities [52], making it similar to our SmartGroup system.

SmartGroup is designed based upon this rich foundation of peer assessment research, but it differs from available tools in the following key ways: (1) peer evaluations are for group work as opposed to individuals' work (e.g., as opposed to PeerStudio [47]) or individuals' contributions (e.g., as opposed to CATME [51]); (2) grades are to be determined entirely by peer reviews based on instructor-provided rubrics (i.e., as opposed to machine grading [47]); (3) if students feel that their group's grade and the peer reviewer's written rationale for said grade are unfair, SmartGroup offers students a novel approach for disputing a grade through an appeal process. The dearth of research specifically regarding technologically assisted peer evaluations for group outcomes is a significant gap in the literature which we hope this work can begin to fill.

### 2.3. Grouping Techniques and Tools

Group learning activities offer varied active learning opportunities, but they have until relatively recently remained without a formal framework for their design, implementation, or evaluation [53]; although a full review of said group learning activity processes is beyond this work, the element of group formation needs to be discussed. Grouping techniques range from random to highly structured and can either be facilitated by tools

or human-derived. Non-random grouping techniques in the literature often rely upon evaluations of students' characteristics or responses (e.g., affinity and creativity indexes [54], Felder–Silverman personality axes [55], and student survey responses [56]); non-random techniques which require such student data thus often require more instructor effort than randomized grouping. Technology-assisted non-random grouping research has explored and demonstrated the efficacy of software at grouping students based upon instructor-provided criteria [56], as well as the efficacy of tools in promoting such characteristics as student creativity and originality [54]. For instance, during the COVID-19 pandemic, breakout rooms on the Zoom platform are a typical function to group students for collaborative activities or discussions [57]. Despite the efficacy of these tools, instructor-selected groupings can facilitate greater interaction and learning for students, so long as a grouping method that appropriately meets a group's intended goals is used; note that certain grouping techniques may be better suited for short-term in-class group work (e.g., latent jigsaw grouping), while other techniques (e.g., Felder–Silverman personality axes groupings) may be better for longer group projects [55]. In addition, assigning students to new teams for each successive team activity (i.e., as opposed to having teams persist longer than half a semester) enables the best learning outcomes and can attenuate problematic team behavior patterns (e.g., interpersonal conflict and free riding) [58].

Despite the benefits of non-random grouping and successive reorganization of teams, existing tools and techniques require instructors to adopt more responsibilities and burdens (conducting surveys, collecting student responses, reorganizing teams based on prior memberships, etc.); such new burdens may lead to instructors refusing or being unable to adopt these novel techniques, just as burdens prevent them from adopting active learning techniques [59]. SmartGroup intends to offer instructors a solution for facilitating quality group work with minimal new instructor burdens, and thus hopefully leading to higher adoption rates. SmartGroup differs from existing tools by offering instructors options either to randomly reshuffle teams or to keep consistent groups as they need for all of a course's SGLAs; this novel approach grants instructors who use SmartGroup the flexibility to implement both short and long group projects of varying complexities. This research thus offers to the field, as well as instructors, a flexible tool for various SGLAs which eliminates the extraneous grouping burdens placed upon both students and instructors.

## 3. SmartGroup System Design

Noting the benefits of both SGLAs and peer assessment, as well as the difficulties in utilizing these techniques to foster said benefits, we design SmartGroup to facilitate both. SmartGroup is an SGLA class management tool, which is separated into two different web-based user interfaces (Figures 1 and 2). The system takes instructor input (a learning activity description, assessment rubric, etc.), creates and manages the small group activity, and then provides the instructor a set of grades based on peer evaluations (Figure 3). SmartGroup does not directly manage whatever software might be used by students during their SGLAs; it only manages who is in the team, collates group assignments, passes the finished group assignments on to peer assessors, delivers feedback and assignment grades to the original student teams, and then finally provides instructors with grades. The goal of the SmartGroup system is to reduce instructors' burdens and increase their abilities to employ novel active learning techniques. To ensure system sustainability and promote ease of adoption, we, therefore, utilized generic components, created simple user interfaces, and integrated our system with as little other software as possible. We describe key design elements, system implementation, and both user interfaces in the following subsections.

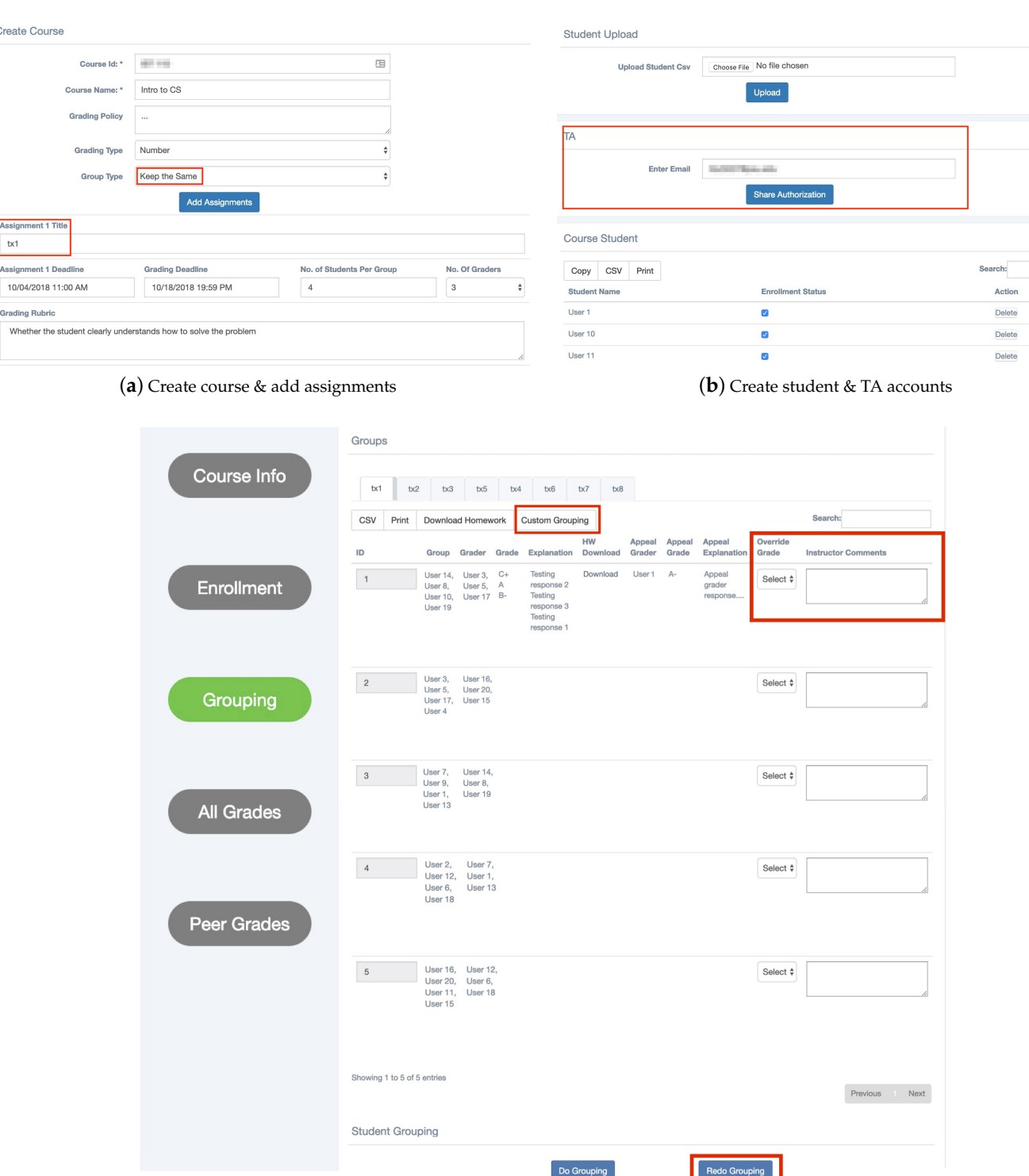

(**a**) Create course & add assignments

(**b**) Create student & TA accounts

(**c**) Grouping & monitoring SGLAs

**Figure 1.** Finalized instructor user interface. Note that system updates are boxed with red.

**(a)** Peer evaluation

**(b)** Upload assignment & tracking group activities

**(c)** Appeal submission

**(d)** Appeal grader

**Figure 2.** Finalized student user interface. Note that system updates are boxed with red.

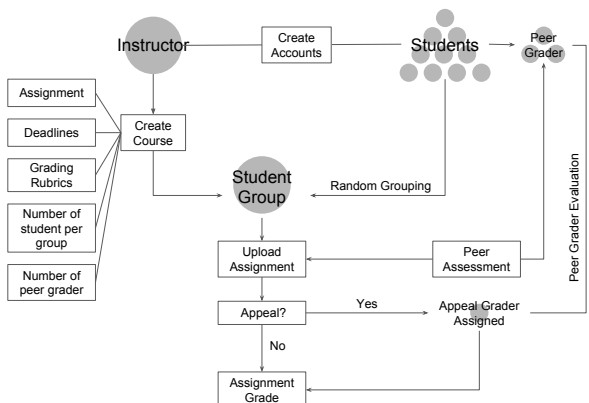

**Figure 3.** Logical flow of how the instructor and students use SmartGroup. Note that human users are represented by grey circles, and actions that require human input are boxed with white backgrounds. Lines denote system elements that facilitate and connect human effort.

*3.1. Design Elements and Rationale*

Prior to building our system, we constructed a framework of rules and goals to guide its design. For example, we decided to have instructors create student accounts (Figure 1b)

to better react to enrollment turbulence before a course, require only one group member to upload the assignment, and provide each team member the same grade and feedback for said assignment, etc. We describe these key design elements in greater detail in the following subsection.

### 3.1.1. Reiterative Grouping

SmartGroup groups students iteratively without prior assessment of their individual skills, strengths, weaknesses, or personality traits. The point of this design idea is not to be random about groups, but to make it so easy to reassign groups that the elaborate analysis and balancing of group membership is not needed. This grouping strategy thus offers lower instructor and student burdens during formation. Successive refactoring of teams (i.e., grouping individuals on prior group membership to avoid successive teams comprised of the same members) was considered, but discarded because small classrooms (i.e., our preliminary deployment course) would not be able to sustain such a strategy. In addition, to avoid that students who are placed into teams based on their personalities or strengths may "perform tasks in the same constellation throughout a longer period" [60], SmartGroup is designed to always use reiterative grouping as the default so that students would not miss out on potential growth opportunities.

### 3.1.2. Reshuffle

Noting the benefits of not keeping consistent teams for a semester, SmartGroup is designed to group students randomly and allows instructors to reshuffle groups (Figure 1a) between successive SGLAs. Effective teamwork depends on students' skills to negotiate authority in small peer groups and manage the possible conflicts [61]. Group shuffling cannot prevent interpersonal conflicts in students' collaboration, but it does minimize the maximum duration of such conflicts within a group by allowing students to change to different teams. We note that reshuffling may make the development of peer group relationships difficult, but the goal of our system is not to form consistent roles for students based on strengths, but to promote exposure to diverse experiences for growth.

### 3.1.3. Grading Rationale

To ease grading burdens, we defaulted SmartGroup to use letter grades (i.e., with the addition of + and −, Figure 1a) instead of numerical grades; students could more intuitively understand that excellent work deserves an A, whereas they might not be able to differentiate nuances to determine if a work deserves an 86 or an 83 [62]. Despite this default setting, SmartGroup is also able to support numerical grading. The instructor must adjust the number of students per group, and the number of peer assessors for each assignment. The grade for the assignment is the average of all peer assessors' grades. Peer assessors are required by the system to provide explanations for their grading rationale for all assignments (Figure 2). This promotes accountability, responsibility, and participation. These explanations also serve as invaluable feedback for the assessees, and the assessor may better internalize assignment expectations by working with a rubric and others' work. This requirement thus offers both assessees and assessors opportunities to grow through either receiving or providing feedback.

### 3.1.4. Peer Assessment

Peer assessment included both peer grading and peer feedback provision in our original iteration. The design decision to include peer assessment was intended for promoting student active learning and reflection [63]; this approach can provide students with quick feedback and grades while reducing instructors' burdens for providing said feedback and grades [64]. This feedback is critical, as it allows students to engage with assignments with which they recently engaged as a learner from the lens of an evaluator; this approach thus promotes meta-cognitive benefits in student active learning. We adopted a double-blind review process for the SmartGroup system. Peer assessors remain anonymous to their

teams. All assessors are randomly assigned and they could not assess their own groups or the groups of those who are assessing their own group. Although we know true anonymity is unlikely in small classrooms, we believe these measures limit unwanted activities (cheating, harsh grading based on revenge, inflation of grades based on friendships, etc.) [65]. SmartGroup provides peer assessors with a great degree of responsibility and authority because we believe that student peer assessment is an opportunity for active learning [63].

### 3.1.5. Appeals and Appeal Assessors

Unique to our SGLA system is the role of an appeal assessor, which essentially works as a mediator for grade disputes. To initiate an appeal, all group members must agree to make an appeal, and one student representative must provide a written rationale for their appeal claim. By submitting this rationale in the student interface, the system sends relevant material (rubric, assignment, peer grades and feedback, appeal argument, etc.) to an appeal assessor (Figure 2d). Appeals can only be made one time per individual project. Considering the significant power appeal assessors carry, we choose them carefully. For our system, we choose appeal assessors from peer assessors who do not have appeals against them; if assessors do not have appeals against them, then they are considered to have provided good peer evaluation by the system. If an appeal is requested, an appeal assessor is assigned to assess the project. The appeal assessor's grade for the project supersedes the average grade (i.e., if there were multiple assessors) originally provided. We intend for appeal assessors to read peer assessors' feedback first (i.e., to ensure that peer assessors' perspectives are considered and assessed, thus limiting the potential negative effects of consolidating authority in appeal assessors), but they may choose to read the material in any order. We are aware that this decision might introduce bias, but we think it is necessary to reduce appeal assessor authority and ensure peer assessors' opinions are heard.

### 3.1.6. Peer Assessor Evaluation

To ensure that students provide high-quality feedback, and thus benefit from higher engagement with rubric requirements and others' perspectives, we designed SmartGroup to provide grades for peer assessors' evaluations. To simplify the process, the system automatically assigns peer evaluators a grade value of 100 for their evaluation unless an appeal is in process for a given evaluation. If an appeal is requested, the appeal assessor is assigned to the assignment and conducts an assessment as discussed above. Once the appeal assessor has assigned a grade, the system automatically checks the deviance of each peer assessor's initial grades from the appeal grade. These values of deviance will be subtracted from the peer assessors' evaluation grades. To clarify, if an appeal assessor grades an assignment as an A, but a peer assessor grades it as a C, then there is a deviance of 7 grade values (i.e., A, A−, B+, B, B−, C+, C); each grade value constitutes 10 points of a peer assessor's evaluation grade, so the peer assessor would receive 30 points for his or her evaluation grade (i.e., $100 - (10 \times 7) = 30$). The minimum value for a peer evaluation grade is set to 0. These evaluation grades are intended for instructor assessment of student performance in peer grading activities and can either be used as a qualitative assessment or as a portion of the student's course grade (in the form of required work, extra credit, etc.); the exact use of these values in a given course is left to instructor discretion, although we believe the potential threat or reward of using these grade values will keep students accountable to produce higher quality assessments and thus benefit more from active learning opportunities.

### 3.2. Implementation and User Interface

SmartGroup is built as a web application in the Python web framework. It uses Django and HTML/CSS for the frontend, Jquery for the backend, Nginx as a web server, and PostgreSQL for the database. GitHub acts as our version control system. The system is split into two user interfaces, one for instructors and the other for students. We will discuss each user interface in more detail in the following sections.

### 3.2.1. Instructor User Interface

Instructors are responsible for three basic requirements when creating a course with SmartGroup: (1) using the "Course Info" tab to create the course (Figure 1a) by providing high-level grading policies (course name and section, grading system choice, assignment grade weight towards final grades, etc.) and assignment information (due dates, assessment deadlines, number of students per group, number of assessors per assignment, SGLA description and rubric, etc.); (2) using the "Enrollment" tab to create student accounts (Figure 1b) by uploading a student enrollment .CSV file from the institution's learning management system (LMS); (3) using the "Grouping" tab to randomly group students for assignments (Figure 1c). The "All grades" tab allows an instructor to export all available grades for group assignments, and the "Peer Grades" tab allows the instructor to view all available peer assessor evaluation grades. These basic factors are necessary for normal lesson plans during course design, so the only significant source of added burden from using this system is transcription. With all of these components are taken care of, the system will create and manage SGLAs and send the instructor a set of grades upon completion. The instructor user interface for managing these steps and for viewing grades can be seen in Figure 1.

### 3.2.2. Student User Interface

The student user interface is designed with a layout similar to the instructor user interface (Figure 2). Students do not need to sign up for an account in the system, as accounts should be created by the instructor (i.e., see Section 3.1). To sign in, the system will thus recognize that a student's account and password are both initially the school account from the .CSV file (i.e., in this case the school-affiliated email address); a password can be changed at any time by the student after his or her initial login. Once logged into their accounts, students are able to see the following series of tabs on the left side of their screens: (1) Your Progress, (2) Group, (3) Peer Evaluation, (4) Grades, (5) Appeal Grader, and (6) Review History. Note that tabs are only active when they have material that students need to take care of, thus negating their need to constantly check every tab. The "Group" tab (Figure 2b) allows students to upload assignments, find group members and their contact information, see deadlines, find assignment grades and peer feedback. Once an assignment has been uploaded, the system chooses a selected number of peer assessors based upon instructor input and allows chosen peer assessors to access the "Peer Evaluation" tab (Figure 2a) to assess the assignment. Students can view a grade for a given assignment by clicking on the "Grades" tab; on the same page, they can also request an appeal by simply putting their arguments into the text box below their grades and comments and clicking the "Appeal" button (Figure 2c). Once an appeal has been requested, the system automatically notifies and sends a consent request to all team members. When consent for appeal is unanimous, the system then assigns an appeal assessor to perform an assessment (Figure 2d); at this point, the "Appeal Grader" tab becomes accessible to the chosen appeal grader. Students can access their previous review history and how their previous peer evaluation is evaluated by the system through the "Review History" tab.

## 4. Methods

SmartGroup employs relatively simple software and user interfaces and, therefore, this study does not represent frontier software design evaluation. Instead, based on findings from a prior study [66], we recognize that one of the key challenges in getting instructors to adopt SGLA technologies is in providing a minimally burdensome system to relieve SGLA management burdens; this is fundamentally a user interface design challenge, and we adopted an iterative design approach. Our goal is thus to begin with a minimally viable system and successively extend and refine the system through walkthroughs, user trials, and field deployments. The current study is the first field deployment. We chose to deploy SmartGroup in a public research university located in the U.S. Mid-Atlantic region for the fall 2018 semester in a course titled "Organization of Data". The iterative design

was performed to meet user needs throughout the deployment; we worked closely with our instructor participant and revised and updated the system based on the instructor's suggestions. This study was approved by the Institutional Review Board (IRB).

*4.1. Study Procedure*

In August 2018, we sought volunteer instructors from the department's faculty mailing list (i.e., approximately 60 members) who met the following eligibility criteria: (1) currently teaching a course that requires students to perform team-based work, and (2) have interest in using the SmartGroup system to manage student group work. We sent our recruiting emails to all faculty members; 2 faculty members replied. We finalized our experiment with one of these members who had decided to utilize group learning in her course plans prior to recruitment. The class design included a ten-week collaboration period with a complex series of team deliverables modeled on the system life cycle; students received feedback on each deliverable. The deployment class had 1 instructor, 1 TA, and 40 students (N = 42). We organized our study procedure into the following three main components: (1) a pre-study introduction for the class; (2) the deployment of SmartGroup; (3) post-study data collection (i.e., survey with students, and interviews with the TA and instructor).

4.1.1. Pre-Study Introduction

A researcher presented the study to the whole class for 15–20 min; this included a presentation of the study goal and procedures. Students were provided with a video tutorial on how to use the system. The instructor allowed the students to deliberate and decide if they wished to be part of the study, and the class unanimously agreed to participate. Students were then offered 3 extra credit points by the instructor after agreeing to participate.

4.1.2. Deployment

The instructor, TA, and students used our system from 4 October 2018 to 14 December 2018, as those were the dates for which the instructor planned to conduct course team-based projects. Students used the system for a total of 7 assignments to upload work and conduct peer evaluations. Students received 3 extra credit points to their grades for using our system. After the experiment, the system was still available for the instructor and students to access. Please note that the 3 extra credits had been awarded to students immediately after they agreed to use the system at the beginning of the class, rather than after they complete the experiment. Therefore, the 3 credit bonus did not influence students' performance and survey responses in the study.

Iterative Design. Our study used iterative design methods [67], and we updated our system based on feedback from the instructor throughout the deployment. The instructor continually communicated system needs and concerns with the primary researcher through email and face-to-face meetings: they would discuss what is needed and why and the primary researcher would then document what had been updated since the initial design. Typically, the primary and secondary researchers addressed each concern within a day to a week.

4.1.3. Post-Study Interview

To evaluate our system we invited all participants to semi-structured interviews, but only the instructor and the TA accepted (n = 2). Students chose not to participate for uncertain reasons, though we suspect that timing (i.e., because invitations were sent during the final exam period) and a lack of compensation may have contributed. Interviews ranged from 60 to 75 min. Interview themes included questions such as: (1) How has SmartGroup enabled, hindered, or changed instruction processes?; (2) What benefits or harm might result from using SmartGroup and peer assessment?; (3) How could SmartGroup be changed or improved to better meet the instructor's needs?; (4) How might the instructor/TA use SmartGroup in the future?

Post-Study Survey. All participants received a link to our anonymous Qualtrics survey. Survey response rates were acceptable (i.e., 47.6%, n = 20). The survey contained 15 questions, and could be finished in 10–15 min. It included 3 clusters of 4–6 Likert-type scale statements about system clarity and ease of use [68], user system satisfaction [69] (see Figure 4), evaluation of functionalities, etc., and 9 open-ended questions which covered themes such as whether there were any problems using the system, avenues for potential improvement, what participants liked and disliked about the system, if participants felt like SmartGroup improved peer assessments, which aspects of the system were most beneficial, and 3 demographic questions (i.e., age, gender, and intended major).

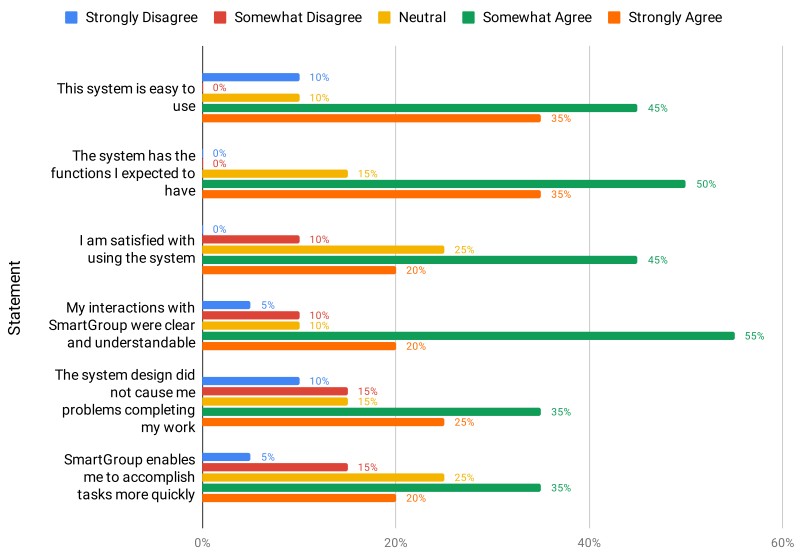

**Figure 4.** Likert-scale user responses regarding system user satisfaction.

### 4.2. Data Analysis

Anonymous survey data were collected and interviews were transcribed from audio recordings. Quantitative data (i.e., Likert scale survey responses) were analyzed using descriptive statistics, and qualitative data (i.e., interviews, open-ended survey responses, meeting notes and correspondence emails) were analyzed using inductive thematic analysis [70]; the primary researcher used open coding and axial coding to identify, analyze, and report patterns within the qualitative data, generating and refining categorical codes using an inductive approach. The resulting high-level themes included peer assessment consequences, iterative system improvements, and design weaknesses.

## 5. SmartGroup Case Study Results

In this section, we report features we improved through iterative system design, peer assessment consequences, and finally participants' suggestions for system improvements.

### 5.1. Iterative Design and Development Process

#### 5.1.1. Keeping Consistent Groups

Maintaining consistent groups during a long project was the first necessary update, as the instructor designed the course to have a substantial final group project. The instructor noted, *"[The] final course project they're putting everything that learn that they have learned a class together to do something that's pretty substantial. In my case they built a database of themselves. They propose a project they do the user studies and propose some use cases for a big database and then they do all the data modeling, database design, implementation and also from an interface a web page using HTML and PHP".* This assignment design requires students to stay within the same group. As a result, we added a "keep the same" group strategy to the system (Figure 1a).

### 5.1.2. Semi-Automated Grouping

Our initial deployment's first grouping assigned 40 students into 10 groups of 4. The instructor manually referenced the performance of each student in the class (i.e., based on individual performances on tests, quizzes, and work performed during the first half of the course) for each group; the instructor found that 9 of the 10 groups showed a good mix of strong, medium, and weak students. The instructor noted, " *I already had some data points about the students performance. They had already taken a test... So I had some data points and I know how students performed but that was individual performance... Some students have better abilities on their own when they're doing their homework, taking a test and so on... I wanted every group to succeed. That's why I was manually checking the random grouping to see if there is any group that may be too weak...".* Furthermore, the instructor noted that several students in the weak group also recognized their group's weaknesses and requested group changes. The instructor then asked us to manually adjust student groups from the backend, because these groups would stay consistent until the end of the course and their group work would account for 30% of their final course grades.

We immediately addressed this problem by adding a function to allow instructors to do multiple reiterative groupings until they were satisfied with the grouping results. However, the instructor did not want to disband the nine satisfactory groups. Instead, we disbanded the one weak group and distributed their members into other groups (i.e., resulting in four groups of five students, and five groups of four) from the backend manually. To allow instructors more control over groupings in the future, we later updated the system to allow them to switch students (Figure 5).

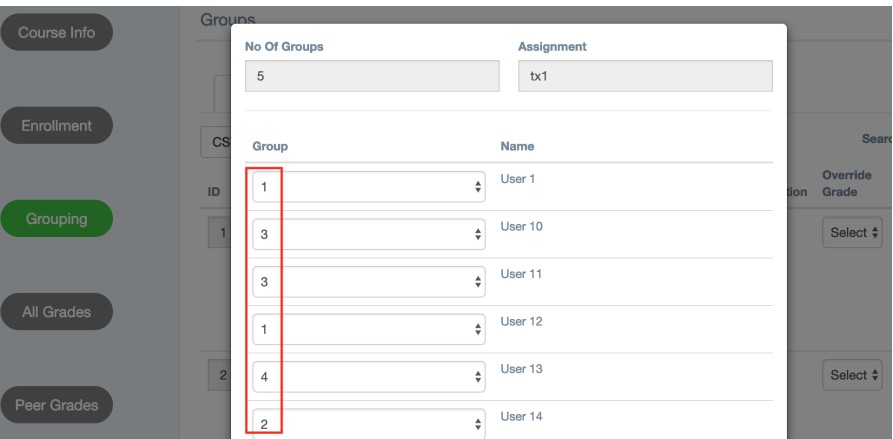

**Figure 5.** Semi-automated grouping. Note that the values in the group column are system defaults from random grouping. Repeated values indicate students share a group. Instructors can manually switch students by changing their group number.

### 5.1.3. Multiple Uploads and Late Submissions

The instructor informed us that certain students would submit a submission before the deadline and then revise and perfect the project until the deadline. However, our original system only allowed students to interact with tabs that required action, and because each assignment had only one input per group, the submission box would disappear after the first submission; the submission box would only reappear after the next work was assigned. The system was designed to allow groups to submit work even after the deadline (i.e., allow for and record late submissions), but because the aim was to simplify and clarify interactions with the system, our submission limit prevented students from submitting revised assignments. To allow students to submit revised works, the instructor devised the following workaround: *"[T]here was one assignment that I added actuaries, because some students wanted to do multiple submission but they were not able to... [so] I created one more assignment so that they can submit the files they wanted to add".* We discussed our rationale with the instructor, and raised our concerns about late submissions. In the end we reached

an agreement that students would be allowed to submit and resubmit an assignment as many times as they desired until one hour after the deadline.

### 5.1.4. Submission Feedback and Group Communication

In addition to challenges submitting multiple assignment versions, the instructor noted that students had difficulties ensuring that assignments were uploaded. We anticipated that students would not worry about assignment uploads if they did not have a tab that required actions, but the opposite was true; as a result of our approach (i.e., only allowing students to click on actionable tabs and only allowing one student per group to upload assignments), students worried that their group assignments had not been uploaded and lacked a means to check. The instructor offered the following suggestion, *"[L]et's say if you give students some feedback after they upload, that they actually have a page to see what they uploaded that will really help them"*. Therefore, we enabled students to see a check mark if their team has submitted the assignment in the "Your Progress" tab. They can also download their submitted files from the "Group" tab (Figure 2b); this allows them to not only see that a file is submitted, but also what version of the assignment file was submitted. In addition, the TA suggested the following during this iteration: *"Maybe the group member they also want to contact each other. So you should also include their email information also in the group section, so that they can easily contact each other at the very beginning"*. We, therefore, displayed team members' contact information in the "Group" tab to allow for easier communication (shown as numbers in Figure 2b).

### 5.1.5. Instructor Grade Overrides

SmartGroup was originally designed to select appeal assessors from peer assessors with good assessment histories (see Section 3.1.5) and automate appeal processes; this was done to minimize instructor burdens and foster student autonomy and growth. Only one group attempted to use this "Appeal" feature; and the instructor describes the situation as follows: *"[The group] did say they [tried] to do the appeal. . . but they kind of told me that. . . they had some problem with going through the appeal process"*. From the backend, we determined that this group failed to get unanimous agreement within the system framework (i.e., not all group members clicked the "agree to appeal" button); therefore, they went outside of the system and asked the instructor directly for an appeal. The instructor investigated the problem and found that the wrong rubric (i.e., for HW5) was used to assess an assignment (i.e., HW6) and decided to override the grade. She contacted us and we provided an updated system with the ability for instructors to manually override grades and provide comments (Figure 1c). This feature may increase instructor burdens from our original design, but it does so minimally (i.e., as an override rather than a necessity) and in a way that provides instructors greater authority in the system.

### 5.1.6. Assignment Title Customization

The SmartGroup system was designed to sequentially title each assignment for ease of use (HW1, HW2, HW3, etc.). However, this led to student confusion. The instructor noted, *"That's why the [assignment] name needs to be clear. That way they see 'oh this is for, for what'. So that needs to be editable. Because otherwise you know, I can communicate to them but they're always students who don't come to class, or students were missed when I said, Yeah. they think they were submitting to the assignment 5, but because assignment five has already ended or close, and the opening is for assignment 6"*. The instructor thus suggested that more concrete or descriptive titles (database design, implementation, etc.) would make student assignment submissions less confusing and thus suggested editable assignment titles; this feature was added after the noted problem with "HW6" (Figure 1a).

### 5.1.7. TA Account and Role

In an attempt to minimize instructor burdens while using the system, our original design did not have an account or active role for teaching assistants (TAs). Because the

system is intended to automate workflow which largely would make a TA redundant, we felt such a role would be unnecessary. However, the course for our deployment study had already had a TA and was only using SmartGroup in the later half of the semester (i.e., the first half of the semester was primarily individual work such as quizzes and tests, and thus could not be managed by SmartGroup). This oversight regarding hybrid (i.e., both group and individual work) teaching techniques created a misalignment between the system design and the broader, unintended instructional stakeholders. During the deployment, the instructor shared her own account with the TA, however, as the TA, noted, *"Actually I didn't use [SmartGroup] too much, because I don't have an account in that system"*. As a result of the TA's concerns, we did create a feature to allow the instructor to "Share Authorization" of the instructor account with the TA (Figure 1b). Essentially, this granted the TA access to the instructor's account with his/her own credentials.

5.1.8. "All" as Assessors for Group Presentations

The final project of our deployment study's course required all members of a group to present their system to their peers. For this presentation, the instructor asked all of the students in the audience to assess the presentation of each presenting group (e.g., if a group of 4 presented, the remaining 36 peers would assess); each group would have to assess eight groups (i.e., a total nine groups). However, such an application would violate SmartGroup's fundamental assessment rules (i.e., double-blind, assessors cannot assess groups which are assessing the assessors). As a result, SmartGroup could not be used for this task, and the instructor thus used paper sheets with five assessment rubrics; the audience assessors would provide grades and feedback on these sheets which would be collected by the TA. The TA was tasked with digitizing the data. Unfortunately, each presentation had 35–36 papers to collect, which roughly resulted in 280 assessment papers. In all of these papers, some students failed to report the team member names or group project name for the presenting group they were evaluating. This problem frustrated the TA, who suggested the following: *"I would suggesting you create some, some function that one group can be evaluated by all the rest of students of the class. That can also save my time and labor because I need to collect all the evaluation, and I need to classify which paper evaluation is for which group"*. We thus created a SmartGroup iteration which allowed the instructor the ability to choose an "All" students option for "No. of Grader" (i.e., number of graders/assessors) on the "add assignment" page (Figure 1a). Once a presenting team had submitted their slides, all peer students, except their own team, would be able to grade these slides and the presentation. This issue occurred at the end of the deployment, so although we did provide these features, they were not tested or used in the deployment itself.

*5.2. Peer Assessment Consequences*

The consequences and benefits of using peer assessment in classrooms are well explored in the literature [41–44,46], but the effects of using peer assessment tools, and stakeholders' perspectives regarding these tools, still need further exploration. Regarding students, the results indicate that our peer assessment feature helps foster potentially beneficial cognitive shifts away from traditional student perspectives, offers students fast, diverse feedback, and helps boost weaker student's course outcomes; however, utilizing student peer grading may also contribute to potentially uniform grades depending upon instructor grading decisions. Regarding instructors and TA, both of them indicate that the SmartGroup reduces burdens in assignment grading.

5.2.1. Cognitive Shifts from Exposure to Diverse Perspectives

Anonymous responses to the open-ended survey questions indicate that respondents perceived and described three categories of peer assessment benefits. Eight respondents believed peer assessment helped them to better understand assignment goals, expectations, and requirements. One respondent stated that peer assessment *"Gave us an approximation for the amount of work required for the assignment"*. Four respondents described peer group

responses and perspectives as beneficial (for comparing and contrasting with works from other groups, understanding what other groups believed was important, etc.). For example, a respondent noted, *"It helped me understand what others saw when they looked at our work and what stood out the most"*. Finally, one respondent noted that the anonymous peer assessment helped to foster awareness of and mitigate biased grading and assessment. The respondent stated, *"It made me see that especially since my name is not on my work, that I should try extra hard the trying and when grading, this mitigating the possible attempt for bias grading towards others"*.

### 5.2.2. Fast, Diverse Feedback

One of the most significant benefits of student peer assessment is fast feedback, which often cannot be provided in large classrooms with large student-to-faculty ratios. As the instructor noted, *"The peer assessment is, first of all, it's very fast, right? The student don't need to wait. Second of all, they get a diverse opinion from multiple graders. And that's good as well. And third of all, you know having having the students see each other's work sometimes inspire them to do more"*. Therefore, the instructor describes how the benefits of fast feedback go beyond just the quickness of feedback and include a diversity of perspectives (i.e., if multiple peer assessors are used) and potentially even motivational influences.

### 5.2.3. Helping Weaker Students Achieve Assignment Parity

Although grade uniformity was a possibility, peer assessment may have also contributed to a grade leveling by sharing ideas and perspectives. The instructor stated, *"I think in the process they should have learned more by looking at other people's work. . . in a way it makes it more homogeneous. . . let's say if you have an individual work, if they don't see what others peoples are doing, then the students or the groups who have really high standards they would do you know really really good. . . But when they start to look at each other, I think [it flattens] the field a little bit. . . I was a little disappointed by a couple of groups because they really have the best people in the class. . . and they didn't do much better than the other groups. . . I felt like they didn't do the extra mile that they could have done. . . So maybe because they see what others are doing. They think 'Oh I am already on the top', Then maybe you know, they have other classes to go to, so. But on the other hand if a group was in general having a lower standards of themselves they see what other groups have done, they will push themselves up"*. She is suggesting that the sharing of perspectives between the students may have removed extremes by both reducing higher achievers' motivation to put in extra effort and increasing lower achievers' motivation to provide deliverables of quality consistent with the rest of the class. Students may be performing cost–benefit analyses on their time as a result of knowing the quality of work being offered by their peers.

### 5.2.4. Cooperative Class Dynamic Increases Student Performance

Despite the benefits of fast feedback, peer assessment may have concerning drawbacks. Most notably, student responses to grading and assessment may be inconsistent, even with rubrics. As the instructor noted, *"The peer grading I think it worked really well. . . But different graders have really different grading criteria. In general the students were really generous to each other. . . I was very specific about what I wanted them to submit, so maybe that helped. And most group[s] did really well. So they definitely deserved it, A minus. . . maybe two or three graders. . . were very strict, like they would give B or even there was once a student who gave a C minus. . . those couple of students looked like outliers. . . everyone else was pretty generous toward each other"*. The potential for grade uniformity, as well as extreme assessment, may be general concerns in small classrooms where anonymity cannot easily be maintained. However, the instructor does note that students deserved high grades, so grade uniformity may potentially represent a benefit of the process; this could be supported by reports from the instructor and TA that the learning environment was predominantly positive (i.e., students interacted positively with others). Even if grades were uniform, though, the instructor is ultimately concerned about student learning rather than student grades. She noted, *"[U]ltimately I think the goal is to make the students learn better and to have the performance the*

*overall class to improve. Because as an instructor myself I want to save some time but I do care a lot about how the students, when and whether the students are learning, and how they are doing. . .”* Therefore, student outcomes may be more important than student grades in this case.

### 5.2.5. Reduce Burdens from Instructor and TA

As mentioned in the design iteration part, the redesign of the SmartGroup is to automate workflow and reduce burdens for the instructor and TA, which has been verified in our interviews with the instructor and TA. In the interview, when asked whether the SmartGroup has reduced grading burden, the instructor said *“Yeah of course, and it’s instant. It’s almost you know very quickly there will be some students who start grading. OK. So the students get some feedback and some grades right away”*. In the same vein, the TA also mentioned that the SmartGroup reduced her burden because *she didn’t need to grade students’ proposal, and their design work*. Other than grading, the SmartGroup also relieves instructors from taking efforts to manage student groups, as the instructor said that *”I was very happy to have this system help me do the grouping, because you know, whatever way you do the grouping, there might be some issues, right? There’s always, there might be some issues, and we ended up being very happy with the group that the system gave us. And then we did a little bit tweaking on top of that.”*

### *5.3. Grouping Consequences*

SmartGroup was originally designed to utilize group reshuffling, but instructor needs regarding designed coursework required us to keep groups consistent throughout SGLAs in the course. We discuss the consequences of our implemented grouping approach below.

### 5.3.1. Group Composition Equalization

SmartGroup was designed to reduce group interpersonal conflicts through reshuffling and reiterative grouping; despite not using the reshuffling, the instructor noted the following about reiterative grouping reducing conflict: *“[I]f I’m telling the student ‘OK, the group is done by the system’, you know it’s like a higher authority. . . It’s fair, and it’s mostly random, and every group has roughly the same number of students. And whoever had concerns with their group, I addressed their concerns with the semi-automated adjusting. . . I think everybody ended up pretty happy with it”*. Here the instructor discussed using her own intuition and expertise to manually (i.e., semi-automated) split a group that she felt was too weak. This decision was ultimately based on the group members’ individual performances. Her decision to split the “weak” group based on individual performances may have worked well for these “weaker” students, who did deliver higher quality deliverables with their new group members, but we note that her expectations for stronger students based on individual performances were not met. She described outcomes as follows: *“[S]ome groups I thought [were] a little bit weak, it turned out that they coordinated really well. . . I was expecting much more from a group with let’s say two or three really strong students, but it turned out their project is not that much better than an average group. . .”*. Therefore, the instructor’s semi-automated grouping may have helped weaker students, but it may not have helped stronger students as much; this indicates an unequal allocation of benefits from the approach.

### 5.3.2. Role Development

The deployment of our current iteration, which utilized components of our original design (e.g., reiterative grouping and requiring only one student per group to submit work) while discarding others (i.e., notably group reshuffling) may have led to unintended role developments during our study. Three respondents reported in our open-ended survey that a single student would consistently submit group assignments. As one respondent noted, *“We had one person submit overtime, as if that was their role in the group dynamic”*. Apparently, the decision to keep groups constant (i.e., instead of reshuffling) may have led to the development of more “permanent” roles within some groups. This role development was not always present, as five respondents note relying on student volunteers, typically chosen during in-person group meetings. Finally, three respondents also described specifically

having the student who felt most comfortable and knowledgeable about an assignment be the one to submit the group's work. Data regarding other potential role development are absent in our dataset.

*5.4. SmartGroup System Evaluation*

In this deployment, SmartGroup was used by 40 students with ages ranging from 18 to 21 (i.e., M = 19.15, SD = 0.81); 60% of respondents were male, and the other 40% were female. In the following sections, we will discuss these participants' responses to system evaluation questions from our survey.

5.4.1. System Usability Evaluation

Most survey respondents agreed that the SmartGroup system was easy to use (i.e., 80% somewhat agree and strongly agree, see Figure 4) and has the function they expected to have (i.e., 85% agreement). In more detail, 85% agreed that messages which appeared in the system were clear and 70% agreed that instructions for commands and choices were clear.

In addition to finding system clarity agreeable, the majority of respondents (i.e., 65%) were also satisfied with using the system. Specifically, 70% enjoyed providing peer assessments, and 85% felt like they had provided reasonable feedback to peers; 70% also felt accountable for delivering high-quality peer assessments. Most respondents (i.e., 80%) believed that providing peer assessments increased their understanding of assignment expectations, and more than half (i.e., 65%) felt that the use of peer assessment enhanced their learning experiences.

Respondents had diverse responses to what they liked most about the SmartGroup system. A total of 40% liked how easy the system was to use, while 35% said that peer assessment (i.e., giving or providing peer feedback) was their favorite aspect. To be more specific, one student noted benefiting from feedback as follows: *"Peer reviews and what others thought made me make changes"*. Another said, *"I enjoyed how [I] can see feedback from graders. I also liked that it makes it easiest to submit group projects when one submission counts for all. Making our group on the website ensures that we all are collaboratively working together"*. This respondent indicated that making groups through the tool and having one submission per group were also beneficial. Notably, three respondents stated that the most beneficial feature of SmartGroup was the collaborative group work; one stated that SmartGroup provides a *"sense of group"*. Finally, one respondent mentioned that the double-blind assessment was the most beneficial aspect, and another discussed liking the table view (Figure 2b) of compiled group data.

5.4.2. Suggested Usability Improvements

Despite the generally positive feedback, survey participants offered valuable suggestions for future SmartGroup iterations; however, not all respondents did provide suggestions. Four respondents were unhappy with the SmartGroup sign-up process (i.e., because students' accounts were created by the instructor by uploading an enrollment .CSV file); likewise, one respondent noted that explanations for using the software could be improved. Two respondents noted that a form of notification (e.g., emails to the group when an assignment has been uploaded or feedback has been provided) could improve interactions with the system. As one respondent noted, *"The lack of general notifications i.e., when someone left a comment when they graded our group's work"*. Interestingly, one respondent suggested using multiple choice assessment instead of rubrics to aid peer reviewers; he or she suggested, *"Make peer review multiple choice so people will actually review [the rubrics]"*. This respondent is suggesting that multiple-choice peer assessments (i.e., as opposed to open-ended assessments based on strict rubrics) might improve peer assessment processes by fostering higher student adherence to assessing duties.

## 6. Discussion

The pre-deployment and post-deployment iterations of the SmartGroup system differ radically from one another, and in ways which provide us with real-world insights that further develop our original design rationale. Notably, our current iteration based on the lessons learned in this deployment better accounts for instructor needs and interests. We are confident that these changes have expanded the potential applications of the SmartGroup system (e.g., by supporting semi-automated grouping, offering instructors more flexibility to control grades, and supporting different grouping methods to be employed based on assignment needs). Furthermore, we identify strengths and weaknesses in the system which can be exploited or reduced, respectively, in future iterations. We will discuss key insights in the following subsections.

### 6.1. Leveling the Field

Our results support the argument that the SmartGroup system has indeed improved students' active learning in a manner consistent with available literature (e.g., see [41] for peer assessment, and [6] for SGLAs); According to the instructors' interview, she mentioned in the interview that the SmartGroup *has created a positive learning environment for students who performed pretty well in group projects, and she would continue using the system for next semester*.

Even though the SmartGroup system was deployed before the COVID-19 pandemic, the pandemic experience and continuing effects on education add to the importance of using SmartGroup to improve students' engagement in learning. SmartGroup facilitates small-group activities where the interactivity between students shall be strengthened that otherwise is lacking in online classroom settings [57]. Moreover, students and faculty will become more used to taking advantage of online tools to assist remote instruction and learning [71] during or after the COVID-19 pandemic, making it even easier for them to adopt the online SmartGroup tool in online classrooms than in physical ones.

Furthermore, our survey participants indicate that the system was generally satisfactory. SmartGroup does appear to foster student outcomes for what the instructor called "weaker" students, as final assignment grades were relatively high, and the projects themselves were all of a similar caliber. The instructor noted being impressed by the performance of even what she considered to be "weaker" students, who out-performed her expectations; this appears to demonstrate SmartGroup's general success at helping her to achieve her goal to foster student growth, despite potentially contributing to grades becoming uniform. Future work determining the trade-offs caused by openness of work sharing in peer group work assessments (i.e., between friendly, open environments with potentially uniform grade distributions and strict, controlled environments with less uniform grades) is warranted.

We note that SmartGroup may not have as concretely helped "stronger" students as it did "weaker" students, as the "stronger" students did not meet the instructor's high expectations. The instructor did not believe that these "stronger" students challenged themselves to produce the best work they could; she suggested these higher-performing "outliers" may have become demotivated as a result of seeing the quality of work produced by peers. We are uncertain as to the exact causes or magnitudes for this phenomenon, if it is real (i.e., we are wary of independent work success being directly correlated to group work expectations, as team success relies upon more than just the general intelligence of team members [72]), but the instructor's suggestion is viable. Furthermore, the small size of the class may have also made accessing all peer groups' works too easy; it is less likely that students would develop a strong feel for what would constitute "enough effort" if they had access to proportionately fewer groups' works. This suspicion is based on studies that suggests that the expected average grade of a course's students is correlated with students' study efforts (e.g., if an A grade is expected as the class average, students will work less) [73]. Another potential explanation is that people who are already learning successfully have much less room for improvement, and any minor improvements demonstrated by

"stronger" students may not have been as easily noticed; this would be akin to the ceiling effect, as good students start out closer to the ceiling. After all, students who are already learning successfully might have less room for improvement. We believe this is a design space that requires greater attention, as we want to foster the growth of all students, even the few high-performing outliers. Future research regarding how to incentivize these higher-performing students who might otherwise be disincentivized to do their best work may be necessary.

*6.2. Fostering Meta-Cognition in Active Learning*

One important current focus in fostering student success through improving learning processes is meta-cognition [74–77]. Meta-cognitive skill development occurs naturally when small groups of students work together, as they learn from each other while learning through providing peer feedback [8]; meta-cognitive learning strategies are thus fostered by combining active learning activities with feedback which provokes self-reflection. Learners employing meta-cognition reflect on and regulate their learning as they are engaged in learning, and we leverage this through SmartGroup's grouping and peer feedback features. SmartGroup supports metacognition in two respects. First, it eases the logistics and instructor burdens of forming groups and employing active learning SGLAs; this provides students with autonomy and accountability to undertake those burdens and actively engage with the material. Because SmartGroup is designed for group work, students will further be incentivized to share understandings and perspectives with peers, thus provoking reflection. Second, peer evaluation causes learners to revisit learning activities in which they were learners, but to experience it as evaluators who provide feedback and guidance to peers. This change of perspective is meta-cognitive because a task that was recently performed is then critically reflected upon. Our results show promising evidence to support that the SmartGroup system fosters meta-cognitive learning opportunities. Notably, survey respondents stated that our system's features helped them to better understand their assignments, others' expectations for those assignments, and even influenced how they themselves graded and provided feedback. Simply put, these respondents note reflecting upon the material and making cognitive changes. Furthermore, the instructor noted that grades were generally high and that assignment submissions were homogeneous; this apparent averaging or equalization might indicate a high degree of cross-group perspective sharing which ultimately resulted in successful final projects.

Self-reflection is key to student meta-cognition, performance, and growth. In our original design, we used written peer feedback to promote reflection and meta-cognition. Our survey participants reported that both providing and receiving peers' feedback contributed to reflection and meta-cognitive activities (see Section 5.4.1). We believe this indicates either: (1) reflection is promoted by receiving and producing feedback in co-located peer assessment, or (2) students subjectively believe that reading feedback contributed to reflection. Students' subjective experiences should not be discounted. Furthermore, students still receive meta-cognitive opportunities from peer assessment when they adopt the assessor role and provide feedback on an assignment they have themselves finished; the literature from workforce contexts supports that reviewing peers' work does improve reviewers' subsequent work [78]. Future work is necessary to determine how best to promote student meta-cognition through SGLA systems, and as such work was beyond the scope of this deployment.

*6.3. Reshaping the Instructor's Activities*

The adoption of new technological supports and SGLAs might shift group management efforts and grading burden to other tasks, notably by altering instructor interaction styles in a way that does not necessarily reduce burden [79]. We noted similar shifts during our deployment regarding grading, feedback provision, and group formation and management. Instructors and TAs have been freed from certain mundane tasks and allowed to spend more time and effort in intellectual and collaborative activities such as assign-

ment planning, rubric design, and coaching students in labs. In addition, we note that the adoption of SGLAs does not necessarily reduce overall instructor burdens; it only alters the types of burdens they experience via altering their styles of interaction (by increasing mediated-learning interactions, through walking around and asking students questions, helping them reach the right conclusions, etc.) [79]. In this regard, the implementation of SmartGroup is consistent with the implementation of SGLAs, as both alter the efforts and burdens of instructors; indeed, the goal of SmartGroup was to reduce several primary learning activity burdens to allow for time and effort to incorporate SGLAs, and we believe it was successful in that regard. However, the adoption of SmartGroup may have unknown indirect effects; we need larger studies in the future to identify the nature and extent of these effects to ultimately allow the field to build better education supporting interventions.

### 6.4. Scalability and Wider Adoption

The SmartGroup system was originally designed to be a minimally burdensome SGLA and peer assessment management tool aimed at reducing instructors' burdens and facilitating active learning teaching opportunities by constructively leveraging students' time and effort. Despite our intentions, our original system design was misaligned with the instructor's needs in this case study. Our intention was to build a scale-free system that could have broader applications by designing it to free instructors' time forming groups and help provide quick feedback and grading; we thus automated these features or provided students with the responsibilities to perform them. However, for more "hands-on" (i.e., our instructor participant's own words) instructors, we learned that tools that shift instructor grading and feedback burdens entirely to students may hinder instructor satisfaction with, and thus adoption of, said tools; if we want our tools to be adopted, we must allow instructors greater flexibility regarding control (semi-automated grouping, overriding grades, etc.) over even the burdens we seek to reduce for them. As a result, many of our iterations during the deployment were geared toward providing instructors with more control over students' grouping and activities (semi-automated grouping based on prior knowledge of students' individual works, grade overrides, etc.). We believe that these changes to allow options regarding the control of said features ultimately allow for more diverse instructors to satisfactorily use the SmartGroup system; in particular, we believe that our system would be able to greatly benefit instructors for massive online open courses (MOOCs), as MOOCs' large class numbers might be ideal for reducing challenges regarding double-blind review anonymity and higher-performing students choosing to underperform based on knowledge of peer work. As of right now, SGLAs are not employed in large online courses such as MOOCs [80,81], despite such courses being able to provide potentially greater diversity of perspectives [82]; therefore, our SmartGroup system could make significant contributions to this area.

### 6.5. Future Opportunities

Several features in the initial SmartGroup iteration were either not used or used in unintended ways during deployment. These include group reshuffling, appeal assessor, and processes. As discussed above, those are important features that need more field deployments and study.

Reshuffling: SmartGroup's initial design was predicated on the idea that group reshuffling would lead to the sharing of diverse ideas and perspectives, reduce the severity of interpersonal conflicts, and prevent role stagnation; however, owing to the design of the course and the needs of the instructor, we could not test the group reshuffle feature in this deployment. Interestingly, we note that three respondents discussed having a student assignment submitter role develop; we are uncertain if these respondents belonged to the same group, but this does suggest that student roles may develop from prolonged consistent groups. Noting that student groups persisted for 10 weeks (i.e., longer than the maximum length for optimal learning [58]), we believe reshuffling may have reduced this development and should be deployed and evaluated fully. Just as we note that peer

assessment provides benefits from both the assessee and assessor roles [46], we believe role stagnation within groups may unequally distribute learning opportunities and benefits; we intended to help promote student growth by ensuring each student experienced a diversity of roles and tasks, so even the limited evidence of role stagnation (i.e., dedicated submitters) we have may be cause for concern. However, examining student role development is outside of our current study's scope (i.e., an instructor's tool for facilitating SGLA adoption). Future dedicated research may be warranted regarding role development from peer group work assessment tool adoption.

Appeal Assessors and Processes: The appeal process was used once during deployment; it was initiated because a peer assessor had mistakenly judged an assignment using the wrong rubric. If not for this human error, the appeal system would not have been used, as we know that most respondents were happy with the quality of received peer assessment feedback. This use of the system's appeal processes was hindered by the group's lack of in-system appeal consensus and resulted in said group going outside of the system to ask the instructor for an appeal directly. We believe the necessity of consensus for the appeal may not have been communicated clearly or often enough for the students to be familiar enough with the system to utilize its features; in the future, we believe this requirement can be communicated more clearly with reminders, especially considering how rarely students may use such processes. We further believe that the scarcity of appeals, independent of survey responses, represents successful feedback delivery and student satisfaction with peer assessments. However, we still believe the appeal assessor role offers novel active learning opportunities despite the scarcity of its use and should be pursued further.

Design for the Benefits of Learners: Looking beyond the specific SmartGroup tool, the future education system designers should consider that the learners themselves might both contribute to and benefit from organizing and evaluating their own learning activities. We believe that this design direction is always available to some extent in developing new educational support systems.

*6.6. Limitations*

Although this preliminary deployment offered valuable insights, expectations should be tempered based on a few limitations. First, this study focuses on one course with a limited number of student participants and one teacher being studied, which may reduce the generability of the study results to other course contexts and student/teacher groups. Future work could recruit a larger number of teachers and students who are from diverse class contexts to examine the effectiveness of the SmartGroup on promoting collaborative learning and alleviating instructors' burdens. More diverse course contexts would also generate interesting insights into the impact of different courses and content on the use of SmartGroup.

Second, even though the survey data shows that students' attitude and experience with the SmartGroup is positive, it still remains to be examined whether SGLAs will result in extra workload for students who are supposed to not only complete assignments but also peer assessments. In this study, since no students came to the office hour to be interviewed, we could not obtain the interview data detailing students' experience with the SGLAs. Future work could verify whether the workload students need to undertake for peer assessment outweigh the gains they get from small group learning activities or not, and if yes, how such costs of SGLAs could be reduced.

Third, survey responses were adequate, but may have been biased towards proactive participants; we may not have captured the entire range of student experiences. Although we observe high and uniform grades, we note this was by the instructor's design (i.e., rubrics and grading choices). This was only our initial deployment, and in order to respond to the instructor's needs, we did not test certain features from our initial design. Although we gained valuable insights into instructor needs for SGLA facilitating systems, further work to account for these limitations is necessary.

## 7. Conclusions

SmartGroup is an SGLA-facilitating tool that reduces instructor group management and assessment burdens while fostering student active learning opportunities; through collaborating with our instructor participant in our iterative design process, we developed a deeper understanding of the design space. The tangible experience resulted in a hybrid semi-automated approach which might be more apt at enabling "hands-on" instructors; continuous iterations were provided during the length of the 10-week deployment to allow the instructor more control over grouping, grading, and other customization features. Key intended features (i.e., group reshuffling and student appeal assessors) were significantly altered or dropped for this deployment, although our results indicate that such features might be beneficial. Post-study interviews and survey results indicated that participants felt that adoption of the SmartGroup system largely benefited students (e.g., exposure to diverse peer perspectives, reflection, and meta-cognitive opportunities improved "weaker" students' outcomes); although some potential negative effects were noted (e.g., uniform grades and knowledge of the average quality of work leading to "stronger" student underachievement). Furthermore, participant responses were predominantly positive regarding usability and system satisfaction. We believe this initial deployment provides sufficient grounds to begin larger and more targeted deployments of the SmartGroup system regarding promoting scale-free SGLAs.

**Author Contributions:** Conceptualization, J.M.C.; Methodology, H.Z. and J.M.C.; Software, N.K.R.; Investigation, H.Z.; Data curation, N.K.R.; Writing—original draft, H.Z.; Writing—review & editing, N.L. and J.M.C.; Supervision, J.M.C. All authors have read and agreed to the published version of the manuscript.

**Funding:** This research received no external funding.

**Data Availability Statement:** Not applicable.

**Conflicts of Interest:** The authors declare no conflict of interest.

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
