# Peer review of "SmartGroup: A Tool for Small-Group Learning Activities"

_futureinternet, doi:10.3390/fi15010007_

Round 1

Reviewer 1 Report

The authors describe a tool to support CSCW in university classes, allowing group work, peer grading and shuffled groups.

The topic is important and in principle interesting. Nevertheless, I have some bigger concerns regarding the submitted paper, so I that I cannot recommend accepting it in the current state.

The major issues I see are:

-          Topicality: The literature review ends 2017. Since then, a lot of research was done in this field which should be discussed in a paper that will appear in 2023. Especially caused by the COVID-10-pandemic a lot of tools and methods were developed and the novelty of the presented approach should take these into account. Overall, I have the impression that this work was performed around 2018 and then not further updated.

-          Structure of the paper: the paper on the one hand describes a didactical method on the other hand a software tool. Of course, they are related to each other, but I would prefer either a clearer focus to only one of these in a paper or a clearer structure in the paper. One solution could maybe be, to start with the didactic part ending with a list or table of requirements for a tool, followed by literature and market analysis describing, which requirements are already fulfilled by existing products (e.g. Moodle supports group work and peer review) and research approaches and then arguing, why a new tool is required (if so, and not an adaptation of existing tools is maybe more efficient and meaningful in an university infrastructure) and then describe the technical innovation based on the requirements. Evaluation should then concentrate more on the novel aspects, which are much more interesting than standard procedures like user enrolment.

-          Decisions in section 3 should have a more solid base. Some are well justified, some are speculative. No reference is provided here.

I hope, the authors can provide a more up-to-date version of the paper, because, as said before, the field of research is important.

Author Response

Reviewer 1

The authors describe a tool to support CSCW in university classes, allowing group work, peer grading and shuffled groups.

The topic is important and in principle interesting. Nevertheless, I have some bigger concerns regarding the submitted paper, so I that I cannot recommend accepting it in the current state.

The major issues I see are:

-          Topicality: The literature review ends 2017. Since then, a lot of research was done in this field which should be discussed in a paper that will appear in 2023. Especially caused by the COVID-10-pandemic a lot of tools and methods were developed and the novelty of the presented approach should take these into account. Overall, I have the impression that this work was performed around 2018 and then not further updated.

We reframed the introduction part by situating the study in the COVID-19 pandemic context.  In the literature review part, we also added some new references after the year of 2018.

-          Structure of the paper: the paper on the one hand describes a didactical method on the other hand a software tool. Of course, they are related to each other, but I would prefer either a clearer focus to only one of these in a paper or a clearer structure in the paper. One solution could maybe be, to start with the didactic part ending with a list or table of requirements for a tool, followed by literature and market analysis describing, which requirements are already fulfilled by existing products (e.g. Moodle supports group work and peer review) and research approaches and then arguing, why a new tool is required (if so, and not an adaptation of existing tools is maybe more efficient and meaningful in an university infrastructure) and then describe the technical innovation based on the requirements. Evaluation should then concentrate more on the novel aspects, which are much more interesting than standard procedures like user enrolment.

As suggested, we reframed our introduction part by starting with the weaknesses of teacher-centered didactic instruction. Then we described how small group learning activities could improve students active learning and compensate the weakness of didactic instruction, as well as how COVID-19 pandemic makes it more necessary to develop a new tool to improve students’ engagement in online learning. Following that, we discuss the existing products for students’ group work, and compare the existing tools with our tool (SmartGroup) to show its novel aspects.

-          Decisions in section 3 should have a more solid base. Some are well justified, some are speculative. No reference is provided here.

 We add some new references in section 3 to provide evidence for our design rationale.

Reviewer 2 Report

Dear Authors, 

Thank you for your work & submission by providing a tool to help foster an easier adoption of SGLA for teachers and limit academic overhead for our limited time. Mainly I am surprised that you only publish this work now, 4 years after the study. While I do generally am happy with the findings outlined, there are many open ends which would be questions for follow-up surveys and studies and for a journal, after 4 years since the initial study, I would have hoped to read about these. 

Now judging the content as if it had not been for this 4 years period. 

You challenge an extremely important area and extremely important questions, as the benefits of SGLA should not be hindered by teacher overhead. Generally, the paper was well written, easy to follow and good-to-understand. I'll outline the room for improvement which I see below and would suggest you to implement according revisions (where applicable) for this publication: 

(1) What bothers me the most is that the main goal of this research, namely, "[...] reduce instructor burdens for adopting this [SGLA] collaborative learning technique" (c.f. Page 2) is - to my understanding - barely answered by as vague statements as "Instructors and TAs should have been freed from certain mundane tasks [...]" (c.f. Page 19). Were they freed or not? You conducted interviews, how can such a crucial question be answered this loosely? The final sentences of your conclusion do not even name the instructors feedback and change of perspective or of time. 

(2) The study has a very narrow participant group and only one course with one teacher was studied. More students, more teachers would allow for comparison and more generalizable feedback. 

(3) Paragraph 3.1.3. "We Believe, students more intuitively understand [...]" I am sure, you'll find related work to foster a more scientific foundation for that decision instead of believing. 

(4) Have you evaluated (at least you didn't write about it in your work at hand) the additional effort the system is for the students? Providing (good) reviews can be a daunting task. How much extra work is that based on the general work for the course? Has the teacher taken this extra work into perspective and relieved e.g. one or the other smaller tasks? Have you taken into account the on-top work if it were for appeals needed to be solved by other students or would not appeals be a perfect job for the TA? 

(5) Your figures have much whitespace to their left and right and are due to their size barely readable on printed papers. Use the left-and-right space to make the figures bigger

(5.1) Figure 4 should be a table

(6) In 6.1 you mention that the SGLA "likely improved student learning and learning outcomes". Is there a scientific notion supporting that hidden somewhere? Have you compared marks of this course iteration in comparison to other course iterations? Have you talked to the instructor about his observations for this question? 

(7) I don't know how the additional credit-points offered to participate in the experiments have impacted students responses. Particularly as despite those credit points none of the students have offered to participate in the in-depth post-interviews. The study feels biased having to include such an incentive for students to participate. 

Unfortunately I do see that a majority of the issues outlined above would require reexamination / reevaluation and possibly further interviews with the study participants. Those are probably hard to create reasonably four years after the study has been conducted. But for some of the above outlined issues I am positive towards a hope that you collected more material which can be evaluated but hasn't been so far. 

Author Response

Reviewer 2

Thank you for your work & submission by providing a tool to help foster an easier adoption of SGLA for teachers and limit academic overhead for our limited time. Mainly I am surprised that you only publish this work now, 4 years after the study. While I do generally am happy with the findings outlined, there are many open ends which would be questions for follow-up surveys and studies and for a journal, after 4 years since the initial study, I would have hoped to read about these.

Now judging the content as if it had not been for this 4 years period.

You challenge an extremely important area and extremely important questions, as the benefits of SGLA should not be hindered by teacher overhead. Generally, the paper was well written, easy to follow and good-to-understand. I'll outline the room for improvement which I see below and would suggest you to implement according revisions (where applicable) for this publication:

  • What bothers me the most is that the main goal of this research, namely, "[...] reduce instructor burdens for adopting this [SGLA] collaborative learning technique" (c.f. Page 2) is - to my understanding - barely answered by as vague statements as "Instructors and TAs should have been freed from certain mundane tasks [...]" (c.f. Page 19). Were they freed or not? You conducted interviews, how can such a crucial question be answered this loosely? The final sentences of your conclusion do not even name the instructor's feedback and change of perspective or of time.

We added a new section (5.2.5) under the findings to explain that instructor and TA’s burden for grading and group management have been reduced because of the SmartGroup tool. On page 19, we also made changes to the original sentence.

(2) The study has a very narrow participant group and only one course with one teacher was studied. More students, more teachers would allow for comparison and more generalizable feedback.

In the limitation section, we acknowledged the narrow participant group as a limitation of our work, and we proposed future direction to improve.

(3) Paragraph 3.1.3. "We Believe, students more intuitively understand [...]" I am sure, you'll find related work to foster a more scientific foundation for that decision instead of believing.

We added new reference to strengthen this.

(4) Have you evaluated (at least you didn't write about it in your work at hand) the additional effort the system is for the students? Providing (good) reviews can be a daunting task. How much extra work is that based on the general work for the course? Has the teacher taken this extra work into perspective and relieved e.g. one or the other smaller tasks? Have you taken into account the on-top work if it were for appeals needed to be solved by other students or would not appeals be a perfect job for the TA?

In the limitation part, we acknowledged this part and proposed future work direction.

(5) Your figures have much whitespace to their left and right and are due to their size barely readable on printed papers. Use the left-and-right space to make the figures bigger

Done

(5.1) Figure 4 should be a table

We changed it to a table

(6) In 6.1 you mention that the SGLA "likely improved student learning and learning outcomes". Is there a scientific notion supporting that hidden somewhere? Have you compared marks of this course iteration in comparison to other course iterations? Have you talked to the instructor about his observations for this question?

We rewrote this part and provided evidence from instructor’s interviews, which show that SmartGroup indeed improved students’ active learning.

(7) I don't know how the additional credit-points offered to participate in the experiments have impacted students responses. Particularly as despite those credit points none of the students have offered to participate in the in-depth post-interviews. The study feels biased having to include such an incentive for students to participate.

We explained this in the method part.

Round 2

Reviewer 1 Report

Thank you for submitting the revised version and considering the comments from the first review phase.
The majority of the concerns have been addressed.

In the introduction, the COVID-19 pandemic is now highlighted. However, the development and testing occurred before the pandemic began. Of course, the new forms of teaching as described have greatly increased the importance of such systems, so this is well suited to the motivational situation. However, the impression that the system was tested in a pandemic online setting should be avoided.

Also, it would be worth discussing whether the results among teachers and learners would be similar after the pandemic experience if the test were conducted today, or whether the experience with online teaching would have resulted in different results here.

Mostly, the Discussion and Conclusion closely follow the developed system. It would be desirable to look at the scientific results in a more generalized way, so that other scientists can better see the transferability into their projects. What should system designers consider when working on similar systems?

Unortunately, there are many limitations in the study, but I assume that all currently available results are shown.

Author Response

In the introduction, the COVID-19 pandemic is now highlighted. However, the development and testing occurred before the pandemic began. Of course, the new forms of teaching as described have greatly increased the importance of such systems, so this is well suited to the motivational situation. However, the impression that the system was tested in a pandemic online setting should be avoided.

Changes: Under the introduction part, at the beginning of the third paragraph, we explained that this research study was done before pandemic, but the pandemic experience and continuing effects on education add to the importance of our work.

Also, it would be worth discussing whether the results among teachers and learners would be similar after the pandemic experience if the test were conducted today, or whether the experience with online teaching would have resulted in different results here.

Changes: In section 6.1 discussion part, we added a short paragraph to explain how pandemic might influence the use of the SmartGroup, and how the SmartGroup might be easier to be adopted in online setting.

Mostly, the Discussion and Conclusion closely follow the developed system. It would be desirable to look at the scientific results in a more generalized way, so that other scientists can better see the transferability into their projects. What should system designers consider when working on similar systems?

Changes: At the end of section 6.5, we added a short paragraph to show a general direction beyond specific SmartGroup so that other projects could also adopt in the future.